



# A two-step method to derive combined Fourier-wavelet spectra from space-time data for studying planetary-scale waves, and its Matlab and Python software (cfw v1.0)

Yosuke Yamazaki

Leibniz Institute of Atmospheric Physics at the University of Rostock

Schlossstraße 6, 18225 Kühlungsborn

**Correspondence:** Y. Yamazaki (yamazaki@iap-kborn.de)

**Abstract.** The combined Fourier-wavelet (CFW) transform is a useful technique to characterize planetary-scale waves, such as tides and traveling planetary waves in the Earth's atmosphere. A CFW spectrum, presented in a time versus period diagram, can be used to identify wave activity that is localized in time, similar to a wavelet spectrum. A CFW spectrum can be obtained for each of eastward- and westward-propagating wave components with different zonal wavenumbers. This paper introduces an easy-to-implement method to derive CFW spectra in two steps. In the first step, the Fourier transform is performed in space (longitude), and time series of the space Fourier coefficients are derived. In the second step, the wavelet transform is performed on these time series, and wavelet coefficients are derived. It is shown that the CFW transform can be easily derived from these wavelet coefficients. The results suggest that existing Fourier and wavelet software can be utilized to derive CFW spectra. Matlab and Python scripts are created and made available at https://igit.iap-kborn.de/yy01/cfw that compute CFW spectra using the wavelet software provided by Torrence and Compo (1998). Some application examples are presented using longitude-time data from atmospheric and geomagnetic-field models.

## 1 Introduction

### 1.1 Background and motivation

The Earth's atmosphere can support various types of planetary-scale waves, which zonally extend around a full circle of latitude. Zonal wavenumber is defined as the number of wave cycles that fit within the latitude circle. As the wave propagates eastward or westward, an oscillation is observed at ground stations. The period of the oscillation depends on the zonal phase velocity and zonal wavenumber of the wave,

$$T = \omega^{-1} = \frac{2\pi R_E}{kC}\cos\phi, \tag{1}$$

where $T$ (in s) is the wave period, $\omega$ (in s$^{-1}$) is the wave frequency, $R_E$ (in m) is the Earth's radius, $k$ is the zonal wavenumber, $C$ (in m s$^{-1}$) is the phase speed, and $\phi$ (in rad) is the latitude.

Examples of planetary-scale waves in the atmosphere include atmospheric tides (Lindzen and Chapman, 1969; Forbes, 1984) and traveling planetary waves (Salby, 1984; Madden, 2007). Solar tides, with primary periods at 24 h and 12 h (called




'diurnal' and 'semidiurnal' tides, respectively), are thermally excited through periodic absorption of solar radiation mainly in the troposphere and stratosphere (Forbes, 1982a, b). Dominant modes are the westward-propagating migrating (or Sun-synchronous) diurnal tide with zonal wavenumber 1 (DW1) and migrating semidiurnal tide with zonal wavenumber 2 (SW2).
Besides, non-migrating (or non-Sun-synchronous) modes are also commonly observed, such as eastward-propagating diurnal tides with zonal wavenumber 3 (DE3) and 2 (DE2) (e.g., Hagan and Forbes, 2002; Forbes et al., 2008; Oberheide et al., 2011). Tides propagate vertically upward from the source region. Their amplitude increases with height due to the reduction of atmospheric density, until dissipation eventually takes place in the mesosphere and lower thermosphere (MLT) and prevents
their further growth. As a result, the wave amplitude is often largest in the MLT region.

Traveling planetary waves have a period longer than a day and shorter than several weeks. Some are interpreted as Rossby normal modes, which are predicted by classical wave theory (e.g., Longuet-Higgins, 1968; Kasahara, 1976). Rossby normal modes are solutions to Laplace's tidal equation in an idealized atmosphere with no dissipation and mean winds, and represent free (or resonant) oscillations of the atmosphere (Forbes et al., 1995b). The oscillations that are most commonly observed in
the MLT region have periods about 5–7 days (Hirota and Hirooka, 1984; Wu et al., 1994; Forbes and Zhang, 2017; Qin et al., 2021b), 9–11 days (Hirooka and Hirota, 1985; Forbes and Zhang, 2015) and 14–16 days (Forbes et al., 1995a; Day et al., 2011). They are all westward-propagating with zonal wavenumber 1, and called quasi-6-day wave (Q6DW), quasi-10-day wave (Q10DW) and quasi-16-day wave (Q16DW), respectively. Also in this category are the westward-propagating quasi-28-day wave (Q28DW) with zonal wavenumber 1 (Zhao et al., 2019), the westward-propagating quasi-4-day wave (Q4DW) with
zonal wavenumber 2 (Ma et al., 2020; Yamazaki et al., 2021) and the westward-propagating quasi-7-day wave (Q7DW) with zonal wavenumber 2 (Pogoreltsev et al., 2002). The westward-propagating quasi-2-day wave (Q2DW) with zonal wavenumber 2–4 is also frequently observed in the MLT region (Wu et al., 1993; Gu et al., 2013; Moudden and Forbes, 2014; He et al., 2021), and is sometimes regarded as manifestation of mixed Rossby-gravity modes (e.g., Salby, 1981a; Salby and Callaghan, 2001). Although theoretical Rossby normal modes and mixed Rossby-gravity modes are westward-propagating, observations
sometimes show eastward propagating components around the same period range (e.g., Palo et al., 2007; McDonald et al., 2011; Pancheva et al., 2018; Huang et al., 2021; Fan et al., 2022). Equatorial Kelvin waves (Matsuno, 1966; Holton and Lindzen, 1968) are equatorially-trapped eastward-propagating waves. At MLT heights, the ultra-fast Kelvin wave (UFKW) with zonal wavenumber 1 and a period of ~3 days is frequently detected (e.g., Lieberman and Riggin, 1997; Forbes et al., 2009; Davis et al., 2012; Gasperini et al., 2015; Yamazaki et al., 2020b).

Neither tides nor traveling planetary waves are stationary. Generally, their amplitude varies with season. Besides, tidal amplitude shows marked day-to-day variability in the MLT region (e.g., Miyoshi and Fujiwara, 2003; Pedatella et al., 2012a; Wang et al., 2021b). This can be attributed to the interaction of tidal waves with the mean flow and other waves (e.g., Chang et al., 2011; Lieberman et al., 2015; Siddiqui et al., 2022) as well as to changes in the source of tides (e.g., Miyoshi, 2006; Siddiqui et al., 2019). Enhanced tidal variability is predicted to occur in the MLT region during sudden stratospheric warming
events (e.g, Fuller-Rowell et al., 2010; Pedatella et al., 2012b; Jin et al., 2012). A sudden stratospheric warming is a large-scale meteorological disturbance, which usually occurs in the winter polar stratosphere (e.g., Butler et al., 2015; Baldwin

 

et al., 2021). It can affect the whole atmosphere including different latitudes and height regions (e.g., Pedatella et al., 2018; Goncharenko et al., 2021).

Traveling planetary waves in the MLT region sometimes show a burst of wave activity that lasts for a few wave cycles. This
can be attributed to changes in the zonal mean state of the atmosphere, which controls propagation conditions, atmospheric instability, and critical layers (e.g., Salby, 1981b, c; Liu et al., 2004; Yue et al., 2012; Gan et al., 2018). Large amplification of traveling planetary waves is sometimes observed following sudden stratospheric warming events (e.g., Sassi et al., 2012; Chandran et al., 2013; Gu et al., 2016; Yamazaki and Matthias, 2019; He et al., 2020; Wang et al., 2021a).

The MLT weather is highly variable due to the presence of various planetary-scale waves and other waves from the lower
layers (e.g., Fuller-Rowell et al., 2008; Jin et al., 2011; Liu, 2014). Understanding wave activity in the MLT region is important because it has a significant impact on the region above, i.e., the ionosphere and thermosphere (IT) (e.g., Liu, 2016; Yiğit and Medvedev, 2015). The IT region is where many space infrastructures operate, and is important for the radio communication between the ground and satellites (Schunk and Sojka, 1996; Moldwin, 2022). Many studies have found wave-like signatures in the IT region that correlate with tidal and traveling planetary wave activity in the MLT region (e.g., Laštovička, 2006;
Immel et al., 2006; Oberheide et al., 2009; Pancheva and Mukhtarov, 2010; Gu et al., 2014; Yamazaki, 2018; Gan et al., 2020; Sobhkhiz-Miandehi et al., 2022). This is the motivation behind the present study, which introduces a simple spectral analysis method to evaluate planetary-scale wave activity.

The main objective of this study is to describe an easy-to-implement method to derive 'wavelet-like' spectra using longitude-time data. The wavelet analysis is a multiresolution analysis technique, which is widely used in research fields including the
Earth science (e.g., Kumar and Foufoula-Georgiou, 1997; Torrence and Compo, 1998). A wavelet transform can be performed on time-series data to derive a 'wavelet spectrum', which is usually presented in a time versus period diagram. The wavelet analysis is useful for identifying wave activity that is localized in time. This feature would also be useful for studying planetary-scale waves in the MLT region, whose amplitude is variable. However, the standard wavelet technique is not directly applicable to longitude-time data, which are required for the characterization of planetary-scale waves. For spectral analysis of longitude-
time data, Hayashi (1971) described a Fourier-based technique. In this study, Hayashi's method is combined with the wavelet technique so that it can detect planetary-scale wave activity that is localized in time.

## 1.2 Fourier-based analysis of space-time data

Hayashi (1971) proposed a Fourier-based spectral analysis method for longitude-time data, which was successfully implemented in later studies (e.g., Mechoso and Hartmann, 1982; Wheeler and Kiladis, 1999; Miyoshi and Fujiwara, 2006; Akmaev
et al., 2008; Sassi et al., 2016). The technique involves two steps. In the first step, the Fourier transform is performed in space (longitude), and time series of the sine and cosine Fourier coefficients are derived. In the second step, the Fourier transform is performed on these time series. Hayashi (1971) clarified how the amplitude and phase of eastward- and westward-propagating waves are related to the Fourier coefficients obtained from the second Fourier transform. What follows is a brief review of the technique of Hayashi (1971).





Assuming that perturbations of an atmospheric parameter $W$ (denoted by $\delta W$) at a fixed latitude can be expressed as the sum of eastward- and westward-propagating components with various zonal wavenumbers $k$ (= 0, 1, 2, ...) and frequencies $\omega$ (>0):

$$\delta W = \sum_k \delta W_k = \sum_k \left( \delta W_k^+ + \delta W_k^- \right), \tag{2}$$

where

$$\delta W_k^+ = \sum_\omega R_{k,\omega}^+ \cos\left( \omega t - k\lambda - \varphi_{k,\omega}^+ \right) \tag{3}$$

represents eastward-propagating components, and

$$\delta W_k^- = \sum_\omega R_{k,\omega}^- \cos\left( \omega t + k\lambda - \varphi_{k,\omega}^- \right) \tag{4}$$

is the westward-propagating counterpart. $t$ and $\lambda$ are time (in s) and longitude (in rad), respectively. $R$ and $\varphi$ are the amplitude and phase of the wave component, respectively, with the superscripts $+$ and $-$ indicating the eastward- and westward-
propagating components, respectively. The above equations can be rearranged, and the component with zonal wavenumber $k$ can be written as

$$\delta W_k = C_k\left(t\right) \cos kx + S_k\left(t\right) \sin kx, \tag{5}$$

with

$$C_k\left(t\right) = \sum_\omega \left( A_{k,\omega} \cos \omega t + B_{k,\omega} \sin \omega t \right) \tag{6}$$

$$S_k\left(t\right) = \sum_\omega \left( a_{k,\omega} \cos \omega t + b_{k,\omega} \sin \omega t \right), \tag{7}$$

where

$$A_{k,\omega} = R_{k,\omega}^+ \cos \varphi_{k,\omega}^+ + R_{k,\omega}^- \cos \varphi_{k,\omega}^- \tag{8}$$

$$B_{k,\omega} = R_{k,\omega}^+ \sin \varphi_{k,\omega}^+ + R_{k,\omega}^- \sin \varphi_{k,\omega}^- \tag{9}$$

$$a_{k,\omega} = -R_{k,\omega}^+ \sin \varphi_{k,\omega}^+ + R_{k,\omega}^- \sin \varphi_{k,\omega}^- \tag{10}$$

$$b_{k,\omega} = R_{k,\omega}^+ \cos \varphi_{k,\omega}^+ - R_{k,\omega}^- \cos \varphi_{k,\omega}^-. \tag{11}$$

Equations (8)–(11) can be further rearranged as follows:

$$R_{k,\omega}^\pm \cos \varphi_{k,\omega}^\pm = \frac{1}{2}\left( A_{k,\omega} \pm b_{k,\omega} \right) \tag{12}$$

$$R_{k,\omega}^\pm \sin \varphi_{k,\omega}^\pm = \frac{1}{2}\left( B_{k,\omega} \mp a_{k,\omega} \right), \tag{13}$$



from which $R$ and $\varphi$ can be derived as:

$$R_{k,\omega}^{\pm} = \frac{1}{2}\sqrt{\left(A_{k,\omega} \pm b_{k,\omega}\right)^2 + \left(B_{k,\omega} \mp a_{k,\omega}\right)^2} \tag{14}$$

$$\varphi_{k,\omega}^{\pm} = \arctan\frac{B_{k,\omega} \mp a_{k,\omega}}{A_{k,\omega} \pm b_{k,\omega}} \tag{15}$$

$R_{k,\omega}^{\pm}$ and $\varphi_{k,\omega}^{\pm}$ can be determined using longitude-time data sampled at a fixed latitude by, first, performing the Fourier transform in longitude to obtain time series of the sine and cosine Fourier coefficients (i.e., $S_k(t)$ and $C_k(t)$) and, then, performing the Fourier transform on $S_k(t)$ and $C_k(t)$ to obtain the sine and cosine Fourier coefficients (i.e., $B_{k,\omega}$, $b_{k,\omega}$, $A_{k,\omega}$ and $a_{k,\omega}$).

## 1.3 Wavelet analysis of time series

The wavelet analysis method considered here is the continuous wavelet transform described by Torrence and Compo (1998). For a given time series $x(t)$, the continuous wavelet transform $X$ is defined as the convolution of $x(t)$ with a wavelet function $\Psi$:

$$X(s,\tau) = \int_{-\infty}^{\infty} x(t)\Psi^*\left(\frac{t-\tau}{s}\right)dt, \tag{16}$$

where $s$ is the scaling factor, representing the extent of dilation or compression of the wavelet, and $\tau$ is the translation factor, representing time shift. $\Psi^*$ is the complex conjugate of $\Psi$. For the present study, the Morlet wavelet is used for $\Psi$. The Morlet wavelet is the product of a complex sinusoid and a Gaussian window. That is,

$$\Psi\left(\frac{t}{s}\right) = \left(\cos\frac{\omega_0 t}{s} + i\sin\frac{\omega_0 t}{s}\right)e^{-\frac{1}{2}\left(\frac{t}{s}\right)^2}. \tag{17}$$

$\omega_0$ is usually set to be 6 to satisfy the admissibility condition (e.g., Farge et al., 1992).

If $x(t)$ is sampled with the sampling interval $\Delta t$,

$$t_n = n\Delta t \tag{18}$$

$$x_n = x(n\Delta t) \tag{19}$$

$$\Psi_{n,s} = \Psi\left(\frac{n\Delta t}{s}\right) \tag{20}$$

where $n = \{0, 1, 2, ..., N-1\}$, and $N$ is the number of points in the time series. The wavelet transform (16) can be approximated as follows:

$$X_{n,s} = X(s,n\Delta t) = \sum_{n'=0}^{N-1} x_{n'}\Psi_{n'-n,s}^* \tag{21}$$

In practical application, the equation (21) is not directly used for the computation of $X$. Instead, the Fourier transforms of $x$ are $\Psi$ are used in light of the convolution theorem. The convolution theorem states that the Fourier transform of a convolution





of two functions is the same as the product of the Fourier transforms of the two functions. The discrete Fourier transform of $x$

is:

$$\hat{x}_m = \mathcal{F}\{x_n\} = \sum_{n=0}^{N-1} x_n e^{-i\frac{mn}{N}}, \tag{22}$$

where $m = \{0, 1, 2, ..., N-1\}$ is the frequency index, and $\mathcal{F}$ is the Fourier transform operator. The Fourier transform of the

Morlet wavelet $\Psi$ is:

$$\hat{\Psi}(s\omega) = H(\omega) e^{-\frac{(s\omega - \omega_0)^2}{2}}, \tag{23}$$

where $H(\omega) = 1$ for $\omega > 0$, and $H(\omega) = 0$ for $\omega \leq 0$. The discrete Fourier transform is

$$\hat{\Psi}_m = \mathcal{F}\{\Psi_n\} = \hat{\Psi}(s\omega_m), \tag{24}$$

where

$$\omega_m = \frac{2\pi m}{N\Delta t} \quad (m \leq \frac{N}{2}) \tag{25}$$

$$\omega_m = -\frac{2\pi m}{N\Delta t} \quad (m > \frac{N}{2}). \tag{26}$$

Based on the convolution theorem, the convolution integral of the two functions is the inverse Fourier transform of the product

of the Fourier transforms of the two functions. Thus, the equation (21) can be written as:

$$X_{n,s} = \mathcal{F}^{-1}\{\hat{x}_m \hat{\Psi}_m\}, \tag{27}$$

where $\mathcal{F}^{-1}$ is the operator for the inverse Fourier transform. Thanks to the fast Fourier transform (FFT) algorithm (e.g., Frigo

and Johnson, 1998), the computation of (27) is much faster than the computation of (21).

A wavelet spectrum can be obtained by plotting the amplitude $|X_{n,s}|$ or power $|X_{n,s}|^2$ of the wavelet transform as a function

of time (i.e., $n\Delta t$) and wave period (or scale $s$). According to Meyers et al. (1993), there is a simple relationship between the

wave period $T$ and Morlet wavelet scale $s$:

$$T = \frac{4\pi}{\omega_0 + \sqrt{\omega_0 + 2}} s. \tag{28}$$

Thus, $T = 1.03 \ s$ for $\omega_0 = 6$.

## 1.4  Combined Fourier wavelet analysis

As described in 1.2, Hayashi's method involves two steps. The first step is the Fourier transform of space-time data in longitude,

and the second step is the Fourier transform of the obtained Fourier coefficients in time. This paper explains how the second step

(Fourier analysis in time) can be replaced by the wavelet analysis. It should be noted that the idea of using the wavelet technique

in space-time analysis itself is not new. For instance, Alexander and Shepherd (2010) used the method of Hayashi (1971) to





determine the amplitude of eastward- and westward-propagating planetary-scale waves with different zonal wavenumbers, and then applied the wavelet analysis to the amplitude time series. Mukhtarov et al. (2010) performed least-squares fits of functions in the form of $R_{k,\omega}\cos(\omega t - k\lambda - \phi_{k,\omega})$ tapered by a Gaussian window. They called their technique 'wavelet-periodogram method'. Kikuchi and Wang (2010) used a 2-dimensional (2-D) wavelet transform to analyze longitude-time data, which

enables to identify wave activity that is localized not only in time but also in space. Kikuchi (2014) introduced a simpler version of the technique called 'combined Fourier-wavelet (CFW) transform', which involves the Fourier transform in longitude and wavelet transform in time. Kikuchi (2014) provided a Fortran software. However, since the main focus of Kikuchi (2014) was on the introduction of the CFW concept, rather than the implementation technique, the application of the CFW technique is still generally challenging for non-Fortran users.

The present study introduces an easy-to-implement method, named 'two-step method', for deriving CFW spectra. It allows to compute CFW spectra using existing software of Fourier and wavelet transforms, which are readily available in many data analysis software such as Matlab.

## 2 Methodology

In Hayashi's method, the wave amplitude is assumed to be constant. In order to taken into account localization of wave activity,

the sinusoids in (3) and (4) are replaced by Gaussian-modulated sinusoids. That is,

$$\delta W_k'^+ = \sum_\omega R_{k,\omega}'^+ e^{-\frac{t^2}{2}} \cos\left(\omega t - k\lambda - \varphi_{k,\omega}'^+\right) \tag{29}$$

and

$$\delta W_k'^- = \sum_\omega R_{k,\omega}'^- e^{-\frac{t^2}{2}} \cos\left(\omega t + k\lambda - \varphi_{k,\omega}'^-\right) \tag{30}$$

Accordingly, (6) and (7) are modified as follows:

$$C_k'(t) = \sum_\omega \left(A_{k,\omega}' e^{-\frac{t^2}{2}} \cos\omega t + B_{k,\omega}' e^{-\frac{t^2}{2}} \sin\omega t\right) \tag{31}$$

$$S_k'(t) = \sum_\omega \left(a_{k,\omega}' e^{-\frac{t^2}{2}} \cos\omega t + b_{k,\omega}' e^{-\frac{t^2}{2}} \sin\omega t\right). \tag{32}$$

In analogy to Hayashi's formulas (8–15), the coefficients $A_{k,\omega}'$, $B_{k,\omega}'$, $a_{k,\omega}'$ and $b_{k,\omega}'$ are related to $R_{k,\omega}'^\pm$ and $\varphi_{k,\omega}'^\pm$ as follows:

$$R_{k,\omega}'^\pm = \frac{1}{2}\sqrt{\left(A_{k,\omega}' \pm b_{k,\omega}'\right)^2 + \left(B_{k,\omega}' \mp a_{k,\omega}'\right)^2} \tag{33}$$

$$\varphi_{k,\omega}'^\pm = \arctan\frac{B_{k,\omega}' \mp a_{k,\omega}'}{A_{k,\omega}' \pm b_{k,\omega}'}. \tag{34}$$

Using (17), equations (31) and (32) can be expressed as:

$$C_k'(t) = \sum_\omega \left(A_{k,\omega}' \Re(\Psi^*) - B_{k,\omega}' \Im(\Psi^*)\right) \tag{35}$$

$$S_k'(t) = \sum_\omega \left(a_{k,\omega}' \Re(\Psi^*) - b_{k,\omega}' \Im(\Psi^*)\right), \tag{36}$$





where $\Re(\Psi^*)$ and $\Im(\Psi^*)$ represent the real and imaginary parts of $\Psi^*$, respectively. Just like $A_{k,\omega}$ and $B_{k,\omega}$ which can be obtained as the cosine and sine coefficients of the Fourier transform of $C_k$ (see (6)), $A'_{k,\omega}$ and $B'_{k,\omega}$ can be obtained as the real and negative imaginary coefficients of the wavelet transform of $C'_k$. Similarly, $a'_{k,\omega}$ and $b'_{k,\omega}$ can be obtained as the real and negative imaginary coefficients of the wavelet transform of $S'_k$.

In summary, the amplitude $R'$ and phase $\varphi'$ of eastward $(+)$ and westward $(-)$ propagating wave components with zonal wavenumber $k$ and frequency $\omega$ can be determined in the following two steps. The first step is the Fourier transform of longitude-time data in longitude, which gives the time series of the cosine and sine Fourier coefficients (i.e., $C'_k(t)$ and $S'_k(t)$). The second step is the wavelet transform of $C'_k(t)$ and $S'_k(t)$ in time. The real part of the wavelet coefficients of $C'_k(t)$ and $S'_k(t)$ gives $A'_{k,\omega}$ and $a'_{k,\omega}$, respectively; and the negative imaginary part of the wavelet coefficients of $C'_k(t)$ and $S'_k(t)$ gives $B'_{k,\omega}$ and $b'_{k,\omega}$, respectively. Once $A'_{k,\omega}$, $B'_{k,\omega}$, $a'_{k,\omega}$ and $b'_{k,\omega}$ are determined, $R'^{\pm}_{k,\omega}$ and $\varphi'^{\pm}_{k,\omega}$ can be derived using (33) and (34).

The implementation of the technique is easy, as it requires only standard Fourier and wavelet tools. Matlab and Python software are created and made available at https://igit.iap-kborn.de/yy01/cfw that compute $R'^{\pm}_{k,\omega}$ and $\varphi'^{\pm}_{k,\omega}$ for input data evenly gridded in time and longitude. For the Fourier analysis, the FFT algorithm is used when there are no missing values in the input data; otherwise, the least-squares method (e.g., Wells et al., 1985) is used, which allow gaps in the input data. The wavelet analysis is based on the software provided by Torrence and Compo (1998), which outputs not only the wavelet transform but also other useful parameters such as the 'cone of influence' and the threshold for the 95% confidence level.

## 3 Application examples

In this section, five examples are presented for the application of the two-step method of the CFW analysis to space-time data. The first example uses synthetic data, for which the exact wave composition is known. In the second, third and fourth examples, longitude-time data from atmospheric models are analyzed to demonstrate that the technique can be used to identify planetary-scale waves in the MLT region. In the last example, longitude-time data from a model of the Earth's magnetic field are analyzed. This is to demonstrate that the technique can be used outside the field of atmospheric science.

For the analysis of atmospheric waves, special attention is paid to sudden stratospheric warming events, where tides and traveling planetary waves in the MLT region often show a large response. The events that are well documented in the literature are selected. In earlier studies, a Fourier-based method is most commonly used for examining planetary-scale wave activity during sudden stratospheric warmings. It is possible to evaluate time variations of tides and traveling planetary waves by applying a 2-D Fourier transform to a short-time segment of data and moving the analysis window in time (e.g., Jin et al., 2012). The advantage of the CFW transform is that the computation of spectra is much faster, because it reduces iterative calculations by the use of the convolution theorem and FFT algorithm (see Section 1.3).

### 3.1 Analysis of synthetic data

A 2-D data matrix is created that mimics longitude-time data containing planetary-scale waves. The data, presented in Figure 1a, consist of two wave components, namely 'wave_A' and 'wave_B', along with noise. The wave_A is westward-propagating





with zonal wavenumber $k$=2 (W2) and the wave_B is eastward-propagating with zonal wavenumber $k$=3 (E3). Notations such as W2 and E3 are used in the remainder of this paper, where 'W' and 'E' denote westward- and eastward-propagating components, respectively, and the number that follows W or E represents the zonal wavenumber $k$.

The amplitude of wave_A is depicted in the upper panel of Figure 1b. It changes between 0 and 1 over time in an arbitrary manner. The period of wave_A also changes over time, as shown in the lower panel of Figure 1b. Also presented in the

lower panel of Figure 1b is the CFW amplitude spectrum for the W2 component, as derived using the two-step method. The white curves indicate the 95% significance level estimated using the method described by Torrence and Compo (1998). The white dashed lines show the cone of influence, outside of which the edge effect may not be negligible. The CFW spectrum successfully identifies spectral peaks at the period of wave_A. The spectral amplitude tends to exceed the significance threshold when the amplitude of wave_A is above 0. Figure 1c is similar to Figure 1b except for wave_B. Again, the CFW spectrum

succeeds to identify the amplitude and period of wave_B.

### 3.2   GAIA simulation: Tides and traveling planetary waves during August–October 2019

There was an Antarctic sudden stratospheric warming in September 2019 (Lim et al., 2020; Rao et al., 2020; Yamazaki et al., 2020a). Although this event is categorized as a 'minor' warming (i.e., no reversal of the zonal mean flow at 10 hPa), it was unusually strong for a Southern-Hemisphere event in various measures (Lim et al., 2021), and its effects were observed at

different layers of the atmosphere (e.g., Goncharenko et al., 2020; Noguchi et al., 2020; Safieddine et al., 2020; Wargan et al., 2020; Yamazaki et al., 2020a). A global simulation of the September 2019 sudden stratospheric warming event was presented by Miyoshi and Yamazaki (2020) based on the whole atmosphere model GAIA. GAIA stands for the Ground-to-Topside Model of Atmosphere and Ionosphere for Aeronomy, and detailed model descriptions can be found in Jin et al. (2011) and Miyoshi et al. (2017). Figure 2a shows the polar stratospheric temperature and zonal mean zonal wind velocity at 60°N at

10 hPa during August–October, as derived from the GAIA model. A rapid increase of the polar temperature in September and concurrent reduction of the zonal mean zonal wind velocity are evident, which indicates the occurrence of the sudden stratospheric warming. Since the model is constrained by the JRA55 reanalysis (Kobayashi et al., 2015) below a height of 40 km, these results strongly reflect the JRA55 predictions.

Figure 2b depicts hourly values of the zonal wind velocity over the equator at an altitude of 100 km as a function of time

and longitude. The zonal wind velocity shows considerable variability within the range of ±200 m/s, which is mostly due to waves generated in the region below 40 km. Figures 2c–2h show CFW spectra of the equatorial zonal wind velocity at 100 km for different wave components.

In Figure 2c, the amplitude of the W1 component at a period $T$ of ∼6 days is enhanced around Days 40–70. Earlier studies found that the amplitude of the Q6DW (W1, $T$∼6 days) during the September 2019 sudden stratospheric warming was un-

usually large compared to its seasonal climatology and had a significant impact on the ionosphere (Lin et al., 2020; Gu et al., 2021; Lee et al., 2021; Ma et al., 2022b; Qin et al., 2021a; Yamazaki et al., 2020a; Miyoshi and Yamazaki, 2020; Mitra et al., 2022). In Figure 2e, there is also a hint of the enhanced Q4DW (W2, $T$∼4 days) and Q7DW (W2, $T$∼7 days) around the same time.





In Figure 2d, the UFKW (E1, $T{\sim}3.5$ days) is seen throughout the period. In Figure 2g, the Q2DW (W3, $T{\sim}2$ days) is seen
at the beginning of August 2019, but its amplitude is below the significance threshold. Their wave activity seems unrelated to
the occurrence of the sudden stratospheric warming. Also, there is no apparent correlation between the sudden stratospheric
warming and tidal activity. The most prominent tidal mode in these figures is DE3 (Figure 2h). The amplitude of DE3 is known
to be largest during August–October (e.g., Akmaev et al., 2008; Zhang et al., 2006).

### 3.3  SD/WACCM-X simulation: Tidal variability during January–February 2009

A 'major' Arctic sudden stratospheric warming occurred in January 2009 (Manney et al., 2009; Harada et al., 2010). Whole
atmosphere simulations of this event were presented by several authors (e.g., Fuller-Rowell et al., 2011; Jin et al., 2012; Sassi
et al., 2013; Pedatella et al., 2014; Siddiqui et al., 2021). Siddiqui et al. (2021) used the Whole Atmosphere Community Climate
Model with thermosphere and ionosphere extension (WACCM-X) (Liu et al., 2018) with specified dynamics (SD), in which
the region below 50 km is constrained by the Modern Era Retrospective Analysis for Research and Applications Version 2
(MERRA-2) (Gelaro et al., 2017). The polar temperature and zonal mean zonal wind velocity at 60°N at 10 hPa derived from
this SD/WACCM-X simulation are plotted in Figure 3a for the period of January–February 2009. The reversal of the zonal
mean flow is seen on Day 23, confirming that this event is a major warming.

Observational studies have found large semidiurnal variations in the ionosphere during the January 2009 sudden stratospheric
warming (Goncharenko et al., 2010a, b; Fejer et al., 2010; Yue et al., 2010). Numerical studies clarified that the semidiurnal
ionospheric variations are due to the enhancement of semidiurnal tides that are generated in the lower atmosphere and propagate
into the ionosphere (Jin et al., 2012; Wang et al., 2014; Pedatella et al., 2014). Figure 3b shows the W2 component of the CFW
spectrum for the zonal wind velocity at 50°N and 110 km. An enhancement of SW2 (W2, $T$=12 h) is clearly visible following
the reversal of the zonal mean flow. By performing the CFW analysis at different latitudes, it is possible to visualize the global
structure of SW2 (Figure 3c). It can be seen from Figure 3c that the amplitude of SW2 increased and decreased in the Northern
and Southern Hemispheres, respectively, during the sudden stratospheric warming. A similar plot is shown in Figure 3d but
for DW1 (W1, $T$=24 h) and at 95 km, where the amplitude of DW1 is largest. The relationship between sudden stratospheric
warmings and DW1 tidal variability was discussed in Siddiqui et al. (2022).

### 3.4  SD/WACCM-X simulation: Traveling planetary waves during January–May 2016

A sudden stratospheric warming that coincides with the spring transition is called a 'final' warming (e.g, Black and McDaniel,
2007; Matthias et al., 2021). Studies have noted that a final warming event is often accompanied by a strong Q10DW (W1,
$T{\sim}10$ days) in the MLT region (Yamazaki and Matthias, 2019; Yu et al., 2019; Yin et al., 2022; Qin et al., 2022). Examples
include the final warming event in March 2016. Figure 4a shows the polar temperature and zonal mean zonal wind velocity at
10 hPa as obtained from the SD/WACCM-X simulation presented by Gasperini et al. (2020). The direction of the zonal mean
flow reversed from eastward to westward on Day 65, and did not turn back eastward until the next winter.

Figure 4b displays daily values of the geopotential height at 0.01 hPa ($\sim$77 km) as a function of time and longitude, where
a westward-propagating wave-like perturbation is visible during the final warming. The W1 and E1 components of the CFW



spectrum obtained from these data are presented in Figures 4c and 4d, respectively. A burst of the Q10DW (W1, $T\sim$10 days) during the final warming can be easily identified in the W1 spectrum (Figure 4c). The height profiles of the amplitude and phase of the Q10DW are depicted in Figures 4e and 4f, respectively, for Day 72. The peak of the amplitude is seen at $\sim$70

km. The downward phase propagation (Figure 4f) is consistent with the upward energy propagation of the Q10DW. The characteristics of the Q10DW during the March 2016 final warming derived with the CFW method are in good agreement with the observations presented by Yamazaki and Matthias (2019) based on a least-squares Fourier technique.

As a brief summary, the results presented in Sections 3.2–3.4 demonstrate that CFW spectra derived using the two-step method described in Section 2 are useful for identifying various types of tides and traveling planetary waves in the MLT region

and their temporal variability. The structures of the planetary-scale waves can be determined by performing the CFW analysis at different latitudes and heights.

### 3.5   CHAOS-7 geomagnetic field model: Secular acceleration during 1997–2022

The Earth's magnetic field consists of contributions from various sources (e.g., Olsen and Stolle, 2012). The main contribution comes from the 'core field' (>95% of the total field), which is caused by electric currents flowing in the Earth's core. Other

contributions include the 'crustal field' due to magnetized rocks in the solid Earth (e.g., Maus et al., 2006; Thébault et al., 2016) and the 'magnetospheric field' resulting from large-scale magnetospheric currents (e.g., Lühr et al., 2017). CHAOS-7 is a data-driven model of the geomagnetic field based on magnetic field measurements on the ground and at satellites (Finlay et al., 2020). The radial component of the core field $B_r$ at the core-mantle boundary is evaluated using CHAOS-7. The second-order time derivative of $B_r$ is called 'secular acceleration' (SA) and is often used in the investigation of short-term variations in the

core field (e.g., Bloxham et al., 2002; Chulliat and Maus, 2014; Chulliat et al., 2015; Aubert and Finlay, 2019). Chulliat et al. (2015) analyzed SA data obtained from an earlier version of the CHAOS model, and examined spatio-temporal characteristics of SA at low latitudes during 2000–2015 using a 2-D Fourier transform method. Their analysis is repeated here but for the extended period 1997–2022 using the CFW technique.

Figure 5a displays monthly values of the SA over the equator as a function of time and longitude. Variations on time scales of

several years are visible. Following Chulliat et al. (2015), SA data within $\pm$15° latitudes are separated into the components that are symmetric and antisymmetric about the equator. The CFW analysis is performed on both the symmetric and antisymmetric components. Figures 5b and 5c present the global CFW spectra (i.e., the average of CFW spectra over the entire data period) for the symmetric and antisymmetric components, respectively. Oscillations with periods of 5–9 years are apparent in the symmetric component (Figures 5b). The antisymmetric component (Figure 5c) is much smaller than the symmetric component.

These results are in agreement with those presented by Chulliat et al. (2015). The cause of these oscillations in SA is still under debate. Gillet et al. (2022) pointed out that the oscillations of the core field around periods of 7 years are consistent with Magneto-Coriolis eigenmodes.

In Figure 5b, the dominant modes are those with zonal wavenumbers $k$=3 and $k = 5$. Interestingly, the global CFW spectrum tends to be symmetric about $k$=0. In other words, the data contain approximately the same amount of eastward- and westward-

propagating waves for a given $k$. Implication is that the observed 5–9-year oscillation of SA is associated with standing waves





(rather than traveling waves). The possible contribution from standing waves is also addressed by Chulliat et al. (2015). A standing wave is equivalent to the superposition of eastward- and westward-propagating waves with the same amplitude. Figure 5d shows that the time evolutions of the eastward- and westward-propagating components are similar for both $k$=3 and $k = 5$, further supporting that the oscillations are mainly due to standing waves. Some earlier studies discussed possible ways
to separate standing waves and traveling waves in the analysis of longitude-time data (e.g., Hayashi, 1977, 1979; Pogoreltsev et al., 2002; Ma et al., 2022a). The implementation of such a technique for the CFW analysis would be a topic for a separate study.

## 4 Summary and Conclusions

This study describes a simple method for deriving combined Fourier-wavelet (CFW) spectra (Kikuchi, 2014) from 2-D
longitude-time data. The method is conceptually similar to that of Hayashi (1971), which first performs the Fourier analysis in longitude, then performs the Fourier analysis in time. In the proposed technique, the Fourier analysis in time is replaced by the wavelet analysis (Torrence and Compo, 1998), which can resolve wave activity localized in time. Briefly, the implementation of the technique involves two steps. In the first step, the Fourier transform is performed in longitude, and time series of the sine and cosine Fourier coefficients are derived. In the second step, the wavelet transform is performed on these time
series, and real and imaginary wavelet coefficients are derived. Using these wavelet coefficients, CFW spectra can be obtained separately for eastward- and westward-propagating wave components with different zonal wavenumbers (see Section 2 for details).

Matlab and Python software for computing CFW spectra using the two-step method are created and made available at https://igit.iap-kborn.de/yy01/cfw. Application examples, based on these CFW software, are presented in Section 3. The re-
sults suggest that the technique can successfully identify tides and traveling planetary waves in the mesosphere and lower thermosphere (MLT) region and their transient response to sudden stratospheric warming events (Sections 3.2–3.4), which in previous studies, were most commonly examined based on a short-time 2-D Fourier method using a moving window. The CFW method has an advantage in that the computation is much faster because it avoids iterative calculations that are necessary for the short-term Fourier method. An application example is also presented for a model of the Earth's magnetic field (Sections
3.5), demonstrating that the proposed CFW technique can also be useful in research areas outside atmospheric science.

*Code and data availability.* Matlab and Python software (cfw v1.0) for computing CFW spectra with the two-step method are available at URL: https://igit.iap-kborn.de/yy01/cfw under the GNU General Public License. They can also be downloaded from the Zenodo website at https://doi.org/10.5281/zenodo.7458051. Matlab wavelet software was provided by C. Torrence and G. Compo under the MIT license, and is available at URL: http://atoc.colorado.edu/research/wavelets/. Python wavelet software was created by Evgeniya Predybaylo and Michael
von Papen based on Torrence and Compo (1998), and is also available at the same URL. The GAIA simulation data used in Section 3.2 are available from GFZ Data Services (https://doi.org/10.5880/GFZ.2.3.2020.004). The SD/WACCM-X simulation data used in Section 3.3 are available from https://data.mendeley.com/datasets/47pnw8pgmk/1. The SD/WACCM-X simulation data used in Section 3.4 are available



from https://doi.org/10.26024/5b58-nc53. The geomagnetic field model CHAOS-7 used in Section 3.5 is available from the DTU Space website (http://www.spacecenter.dk/files/magnetic-models/CHAOS-7/).

*Author contributions.* YY was in charge of conceptualizing the study, data analysis, visualization of the results, writing the manuscript, and creating the Matlab and Python scripts.

*Competing interests.* The author declares that he has no conflict of interest.

*Acknowledgements.* The author was supported by the Deutsche Forschungsgemeinschaft (DFG) grant YA-574-3-1.



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



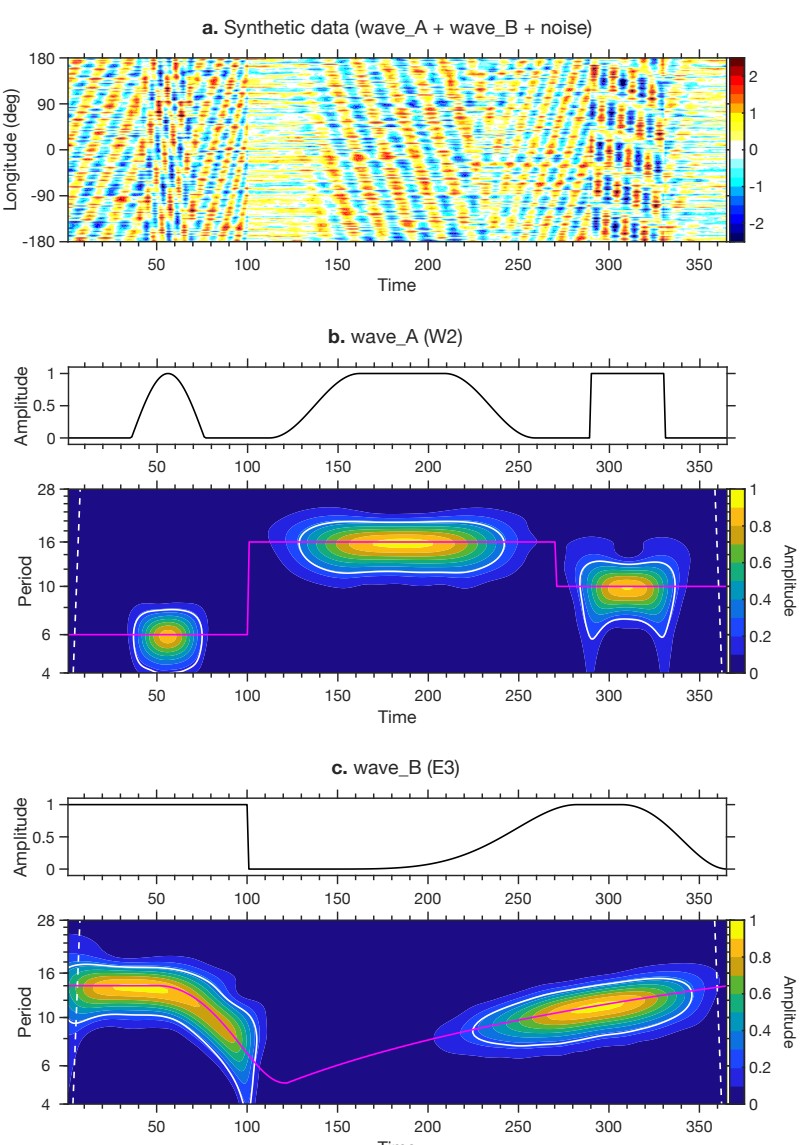

**Figure 1.** (a) Synthetic data containing wave_A (westward-propagating with zonal wavenumber 2, W2) and wave_B (eastward-propagating with zonal wavenumber 3, E3), along with noise. (b) Amplitude and phase of wave_A. The upper panel shows the amplitude, and the lower panel shows the phase (magenta line). The lower panel also shows the combined Fourier-wavelet (CFW) amplitude spectrum for W2. The white curves indicate the 95% confidence level, while the white dashed lines show the cone of influence. (c) Same as (b) except for wave_B. The lower panel shows the CFW spectrum for E3.



**Figure 2.** GAIA model simulation for the period August–October 2019. (a) Polar temperature and zonal mean zonal wind velocity at 60°N at 10 hPa. (b) Zonal wind velocity over the equator at a height of 100 km. (c–h) Combined Fourier-wavelet (CFW) spectra of the equatorial zonal wind velocity at 100 km for (c) the westward-propagating zonal wavenumber 1 (W1) component, (d) the eastward-propagating zonal wavenumber 1 (E1) component, (e) the westward-propagating zonal wavenumber 2 (W2) component, (f) the eastward-propagating zonal wavenumber 2 (E2) component, (g) the westward-propagating zonal wavenumber 3 (W3) component and (h) the eastward-propagating zonal wavenumber 3 (E3) component. The white curves indicate the 95% confidence level, while the white dashed lines show the cone of influence.



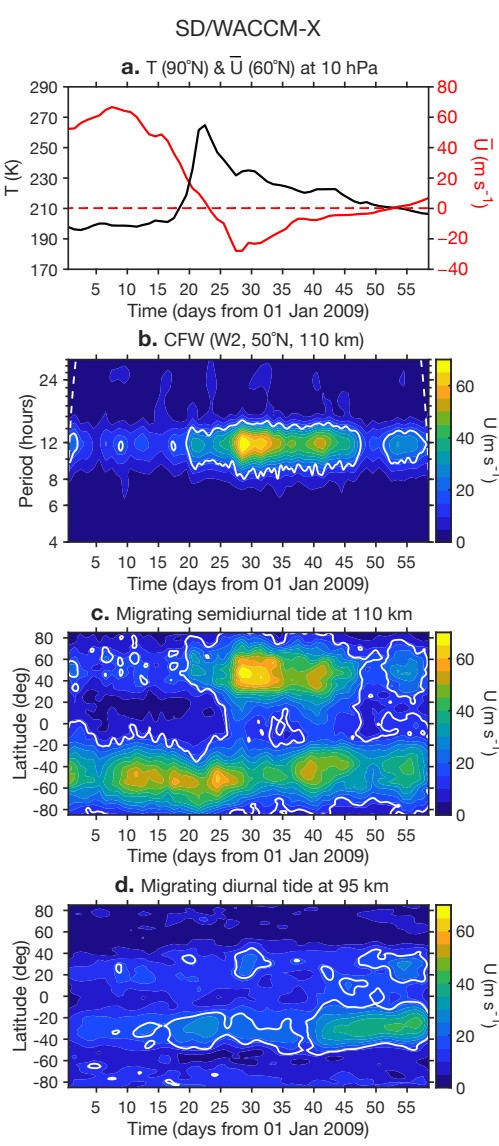

**Figure 3.** SD/WACCM-X model simulation for the period January–February 2009. (a) Polar temperature and zonal mean zonal wind velocity at 60°N at 10 hPa. (b) Combined Fourier-wavelet (CFW) spectrum of the zonal wind velocity at 50°N and 110 km. The white curves indicate the 95% confidence level, while the white dashed lines show the cone of influence. (c) Amplitude of the migrating semidiurnal tide in the zonal wind velocity at 110 km as determined by the CFW technique. (d) Amplitude of the migrating diurnal tide in the zonal wind velocity at 95 km as determined by the CFW technique.





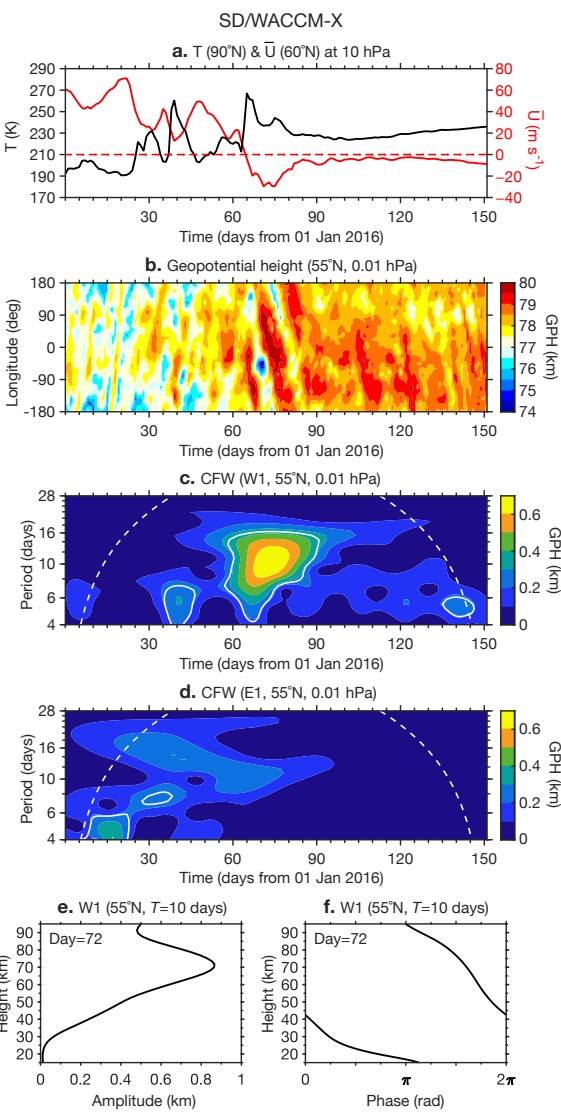

**Figure 4.** SD/WACCM-X model simulation for the period January–May 2016. (a) Polar temperature and zonal mean zonal wind velocity at 60°N at 10 hPa. (b) Geopotential height at 55°N at 0.01 hPa. (c–d) Combined Fourier-wavelet (CFW) spectra of the geopotential height at 55°N at 0.01 hPa for (c) the westward-propagating zonal wavenumber 1 (W1) component and (d) the eastward-propagating zonal wavenumber 1 (E1) component. (e–f) Height profiles of (e) amplitude and (f) phase of the W1 component at a period of 10 days at 55°N and 0.01 hPa on Day 72 as determined by the CFW technique.



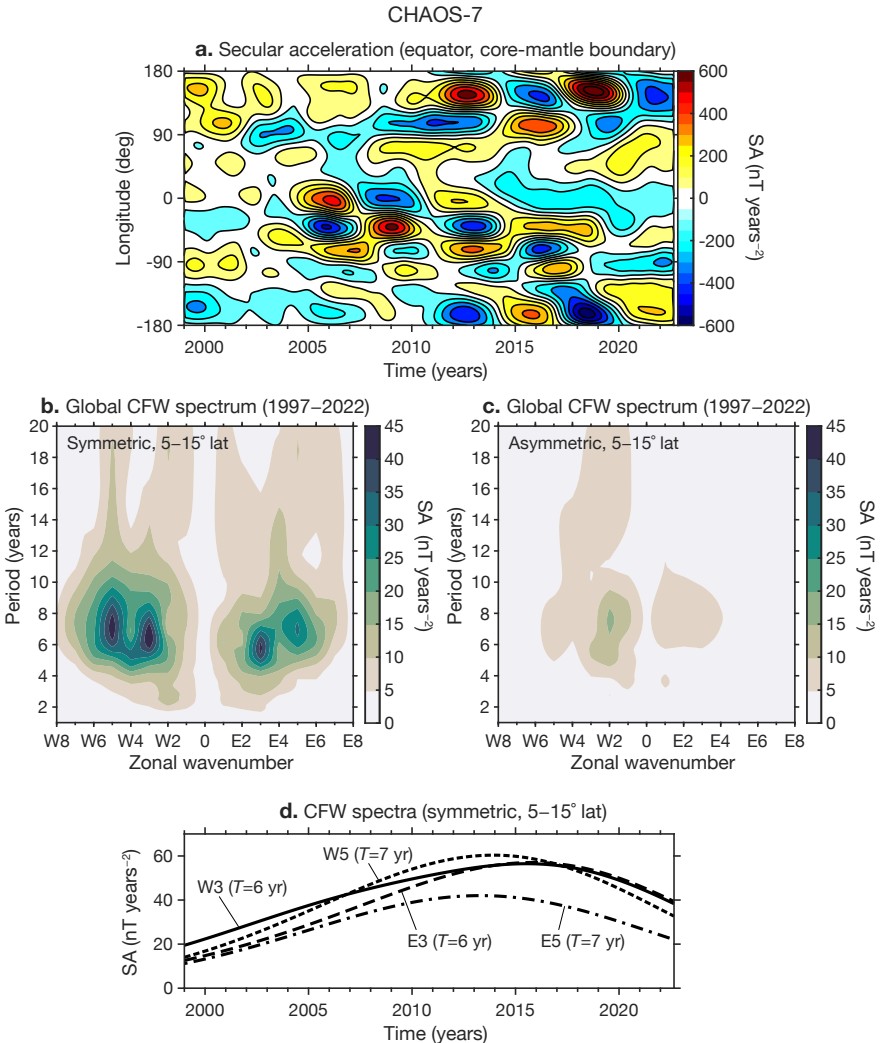

**Figure 5.** (a) CHAOS-7 model prediction of the secular acceleration (SA) in the radial component of the geomagnetic field at the core-mantle boundary over the equator for the period 1997–2022. (b) Global combined Fourier-wavelet (CFW) spectrum for the equatorially symmetric part of SA for 1997–2022. (c) Same as (b) but for the equatorially asymmetric part of SA. (d) Amplitude of the eastward- and westward-propagating zonal wavenumber 3 components (E3 and W3, respectively) of SA at a period of 6 years, and amplitude of the eastward- and westward-propagating zonal wavenumber 5 components (E5 and W5, respectively) of SA at a period of 7 years as determined by the CFW technique.