# Peer review of "A method to derive Fourier-wavelet spectra for the characterization of global-scale waves in the mesosphere and lower thermosphere, and its Matlab and Python software (fourierwavelet v1.1)"

_EGUsphere, 2022_

## Author Comment (AC1)

<h1 style="text-align:center">Replies to Reviewers' comments on</h1>

[previous title] "A two-step method to derive combined Fourier-wavelet spectra from space-time data for studying planetary-scale waves, and its Matlab and Python software (cfw v1.0)"

I am thankful to the reviewers for taking the time to review the manuscript and provide useful comments. I have taken all the comments into account in revising the manuscript. My response to individual comments can be found below, where the comments from the reviewers are highlighted in blue, and the text from the revised manuscript is highlighted in red. The line numbers in the response refer to those in the revised manuscript.

Before going into my replies to individual comments, I would like to highlight important changes that both reviewers may want to be informed of.

[1] The title of the manuscript has been changed. This is mainly because RC1 pointed out that the term "combined Fourier-wavelet" is a misnomer. The new title reads:

"A method to derive Fourier-wavelet spectra for the characterization of global-scale waves in the mesosphere and lower thermosphere, and its Matlab and Python software (fourierwavelet v1.1)"

In the revised manuscript, the technique is referred to as Fourier-wavelet analysis. Accordingly, the name of the software has been changed from "cfw" to "fourierwavelet".

The new title also emphasizes that the main target is global-scale waves in the mesosphere and lower thermosphere.

[2] Figure 1 has been updated by including the results obtained by the least-squares fitting method for a comparison. This is as suggested by RC2. Also, Figure 5 and the relevant text have been removed, as RC1 pointed out that the figure does not take any advantage of the Fourier-wavelet technique.

[Figure]

New Figure 1: "(a) Synthetic data containing wave_A (westward-propagating with zonal wavenumber 2, W2) and wave_B (eastward-propagating with zonal

wavenumber 3, E3), along with noise. (b) Amplitude of wave_A. (c) Phase of wave_A (magenta line), and Fourier-wavelet amplitude spectrum for W2 (contour plot). The white curves indicate the 95% confidence level, while the white dashed lines show the cone of influence. (d) Same as (c) except that the amplitude spectrum is derived with the least-squares fitting method. (e−g) Same as (b−d) but for wave_B. The amplitude spectra are for E3.”

**RC1 (Dr. Jun-Ichi Yano)**

[1-1] This is an interesting piece of work, potentially worthwhile for a publication: there are already extensive literature performing the continuous wavelet transform to time series in atmospheric science. However, this work is new: the continuous wavelet transform is applied to time series, that itself is nothing new, but in the wavenumber space. The author shows that this methodology can characterized certain wave activities rather well, as shown in Figs. 2-4.

[Response] Thank you for accurately articulating the novelty of this work. Below are my replies to individual comments.

[1-2] The main problem with the present manuscript, as it stands for now, is to present it with a rather sensational fanfare, calling it specifically a "combined Fourier-wavelet spectrum". As far as I can follow, it is nothing other than just performing a continuous wavelet transform in time and a Fourier transform in space (longitude): these are two independent linear operations, that can be performed in any order. There is nothing to "combine", but just to perform two independent things in sequence. In other words, the adjective "combined" is nothing other than just a misnomer.

[Response] Output spectra, which are previously called "combined Fourier-wavelet" spectra, are now referred to as "Fourier-wavelet" spectra without "combined". Accordingly, the title of the paper and the name of the software have been changed as stated above.

[1-3] With an attempt of the author convincing the readers that this is "revolutionary", the author provides rather lengthy technical details of the so-called Hayashi's method (that itself is nothing other than just performing the Fourier transform in time and longitude) and the continuous wavelet in the last half of the introduction section. However, I do not see those technical details to be any importance for this method. As I just said above, what is done in practice is very simple: perform FFT in longitude, and the continuous wavelet analysis in time. We just need few more extra words on specific.
On the other hand, the presentation of the proposed methodology in Sec. 2 is rather "muddled", and difficult to get this very simple point straight.

[Response] I do not completely agree with the reviewer. The main idea of this work is to combine the Hayashi's method with the wavelet technique (as mentioned at lines 82-84). Thus, detailed explanation of Hayashi's method in Sec. 1.2 is necessary. As stated at lines 90-91, Sec. 1.2 describes how the amplitude and phase of individual wave components are associated with the 2-D

Fourier transform. Once Sec. 1.2 is understood, Sec. 2. is easy to follow because there are a lot of similarities between the formulae for the Fourier-wavelet method (Sec. 2) and Hayashi's method (Sec. 1.2). No changes have been made to the manuscript.

[1-4] A simple methodology is always a beauty. However, if the author intends to present the present manuscript as a proposal paper of a new methodology, a more careful review of the existing methodologies is required.

First of all, planetary waves can be extracted in a straightforward manner, at least in principle, by the normal-mode decomposition. A full description of this methodology is provided in Zager et al. (2015, GMD, https://gmd.copernicus.org/articles/8/1169/2015/) with a software publicly available to apply this methodology. Also please refer to a workshop report for further backgrounds: Zagar et al. (2016, BAMS, https://journals.ametsoc.org/view/journals/bams/97/6/bams-d-15- 00325.1.xml).

In this respect, the introduction is slightly confused as it stands for now: its

[**Response**] Thank you for the interesting reference. Although normal modes are addressed in Sec. 1, it is not the aim of this paper to identify the normal modes of linear wave theory as in Zager et al. The present study focuses on global-scale waves in the mesosphere and lower thermosphere (MLT) region, where the behavior of traveling planetary waves deviate considerably from classical normal modes. In the revised manuscript, I have emphasized these two facts: (1) the Fourier-wavelet method has been developed for global-scale waves in the MLT region and (2) the behavior of traveling planetary waves in the MLT region deviate considerably from classical normal modes.

For (1), the new title explicitly state "in the mesosphere and lower thermosphere" and the first line of abstract now reads, (lines 1-2) "This paper describes a simple method for characterizing global-scale waves in the mesosphere and lower thermosphere (MLT), such as tides and traveling planetary waves, using two-dimensional longitude-time data."

For (2), it is stated in the revised manuscript (lines 34) "The behavior of traveling planetary waves in the real atmosphere deviates from what is anticipated from the linear wave theory due to dissipation and mean winds."

[1-5] Second is lack of a proper review of the wavelet method. The most important question, in this context, for me is a choice between the continuous redundant wavelet and the discrete orthogonal wavelet. Here, the author chooses the former, but without explanation. The choice is just puzzling for me considering a very fact that the latter is much more robust, with much more potential applicabilities, as my series of work suggest: see a list of reference below.

Performing a continuous wavelet analysis is like a decomposition of a finite domain data into continuous wavenumbers, when only the discrete integer wavenumbers have a meaning.

[**Response**] The advantage of the continuous wavelet transform has been addressed in the revised manuscript

(lines 128-131): "One advantage of the 'continuous' (in contrast to 'discrete') wavelet transform is that the user can arbitrarily select the frequency resolution of the output spectrum. This is helpful especially in investigating traveling planetary waves, as the user has no prior knowledge of the dominant frequency of the wave. On the other hand, the discrete wavelet transform has its own advantage such as non-redundancy and straightforward invertibility (e.g., Yano et al., 2001, 2004), which, however, will not be explored in this study."

Thank you for the references. The Yano et al. (2001, 2004) papers have been cited.

[1-6] Finally, if detection of a planetary-wave packet is the main issue, a discrete set of wavelets can be constructed based on normal modes, in an equivalent manner as the Meyer wavelet is constructed based on the Fourier modes. Though I do not think that the author has to try this possibility in the present work, all those potential possibilities must be clearly mentioned in the manuscript.

It is obvious that the author is only taking a small first step forwards for exploring all those wider possibilities.

[**Response**] Thank you for your understanding. As the reviewer accurately pointed out, this work is just a small first step towards a more comprehensive diagnostic tool for MLT wave dynamics.

The detection of planetary-wave packets is important for this study. In the revised manuscript, I have made it clear that the technique is intended for characterizing global-scale waves in the MLT region (lines 1-3) and that the behavior of traveling planetary waves in the MLT region deviates from that of linear wave theory (lines 34-35). I believe that these changes partly address the reviewer's comment. Also, general future directions are mentioned in Sec. 4, which addresses the rest of the reviewer's comment:

(lines 337-339) "Although this study has focused on tides and traveling planetary waves in the MLT region, the Fourier-wavelet method can be easily applied to data from other regions of the atmosphere. Also, the applicability of the technique in research areas outside atmospheric science is yet to be explored."

Specifics

[1-7] L79-80, the standard wavelet technique is not directly applicable to longitude-time data: false. 2D wavelet transform can easily performed in analogy with the 2D Fourier transform: refer to my publications below.

[**Response**] This was about 1D wavelet transform. The manuscript has been revised to make it clear:

(lines 80-81): "However, the standard 1-D wavelet technique is not directly applicable to two-dimensional (2-D) longitude-time data ..."

 L90, parameter -> variable

[**Response**] The change has been made as suggested.

[1-9] Eqs. (5)-: the frequency, omega, must be discrete, as the case for the wavenumbers. please comment on this

[**Response**] Yes, omega is discrete. This has been made clear:

(lines 94-95) "$\omega$ (>0; $= \omega_0, \omega_1, \omega_2, ...$)"

[1-10] L122: state explicitly that continuous wavelet is applied in time

[**Response**] The text now clearly states so:

(line 125) "A continuous wavelet analysis is applied in time."

[1-11] Eq. (16): the actual data set only has a finite length in time. comment on this

Eq. (21): if not, this expression is puzzling: since continuous wavelet is applied here, obtained coefficients must also be continuous: why we suddenly get a discrete expression?

[**Response**] Yes, the data set has a finite length in time and sampled at a finite rate. That is the reason for the discrete expression. The text has been revised to make this clear:

(line 140) "If x(t) is sampled with the sampling interval $\Delta t$ for a finite length in time from $t_0$ to $t_{N-1}$,"

[1-12] Sec. 2: as far as I can follow, the longitudinal dependence does not play any role in the presentation, though Eqs. (29) and (30) retain it. at least a word would be required for a clarification: otherwise, in my own reading, Sec. 2 is essentially just repeating Sec. 1.3. If not, what is a difference except for a longitudinal dependence added in Eqs. (29) and (30)?

[**Response**] As already described in the text, Eqs (29) and (30) [Eqs (31) and (32) in the revised manuscript] are modifications of Eqs (3) and (4), respectively, taking into account the modulation of the wave amplitude with time. The longitudinal dependence of Eqs (29) and (30) plays an important role in constraining the zonal wavenumber and wave phase. There is no repetition of Sec. 1.3 in Sec. 2. So, no changes have been made to the manuscript.

[1-13] Eqs. (29), (30): the exponent, -t^2/2 must be replaced by -t^2/2s? if not, I do not know how to connect this expansion with (17), as invoked after Eq. (34).

[**Response**] Thank you for spotting the error. Eqs (29) and (30) [Eqs (31) and (32) in the revised manuscript] have been revised by including *s*.

[1-14] Eqs. (29), (30): the given decomposition modes are only localized in time, thus it appears to me that the author essentially fails to address a question of the propagation of a wave packet, that should happen both in time and space.

[Response] As this study is concerned only with temporally-localized global-scale waves, the Fourier-wavelet technique does not take into account the propagation of a longitudinally-localized wave packet in zonal direction. A 'wavelet-wavelet' technique would be required for resolving longitudinally-localized waves. I believe that it is already well emphasized in the manuscript that this paper focuses on global-scale waves. So, no changes have been made.

[1-15] Eqs. (35) and (36): Psi* here must depend on both s and omega: how do you specify them?

[Response] Yes, Psi* depend on both $s$ and $\omega$. The $\omega$ values are given in Eqs (27) and (28), but there was no description on how the scale $s$ was determined. In the revised manuscript, I have explained how the scale $s$ is handled:

(lines 144-149):
"The scaling factor s can be arbitrarily selected. Torrence and Compo (1998) used a set of scales that is fractional powers of two, and it is also adopted here. That is,

$$s_j = s_0 2^{j\Delta j} \qquad\qquad (21)$$

where $s_0 = 2\Delta t$ and $j = \{0, 1, 2, ..., J\}$. $\Delta j$ controls the scale resolution, which the user can arbitrarily select. $J$ determines the largest scale and is given by

$$J = 1/\Delta j \log_2 (N/2) \qquad\qquad (22)"$$

[1-16] Fig. 5b, c: they should be better presented by standard Fourier transforms: the plots do not take any advantage of wavelet, either.

[Response] I agree that Fig. 5 does not take any advantage of the Fourier-wavelet analysis. As such, the figure has been removed along with the relevant text.

**RC2**

[2-1] This paper introduces a simple method to perform combined Fourier-wavelet (CFW) transform to extract planetary-scale waves in the gridded 2-D longitude-time atmospheric data set. Although the concept of this method is not new, or the idea is not difficult to understand, the implementation of this method has always been a difficulty for the space physics or mesosphere and lower thermosphere community. After testing the programs provided by the authors, I think this method has the following advantages:

1. The calculation speed is fast since this method only reads the data once. This advantage is meaningful, especially to the extraction of tides, which usually

requires data with a high temporal resolution and is a challenge for the memory and computing resources of a personal computer;

2. The output is directly the real amplitudes of the waves, which is similar to the S-transform, but this method obtains the amplitudes in the 2-D data set;

3. In fact, there has been no good way to accurately extract the planetary wave activity during sudden stratospheric warmings in the longitude-time satellite observations or simulations. However, the results in this manuscript show that this method does a relatively good job on this issue, which is important for related research.

[**Response**] Thank you for the accurate summary of the advantage of the Fourier-wavelet analysis.

[2-2] Overall, the authors implemented the CFW transform in a relatively simple way with MATLAB and Python, which is very worthy of recognition and will bring a lot of convenience to the middle and upper atmosphere community. I only have two minor comments on the current manuscript:

1. The authors may consider comparing the results obtained by different methods (e.g., CFW transform, 2-D fast Fourier transform, and least squares fitting) to better demonstrate the superiority of the CFW transform in studying the temporal variations of planetary-scale waves.

[**Response**] Figure 1 has been revised by including a comparison with the least-squares fitting method, which is most simple and arguably most popular in studies of global-scale waves in the MLT region. The comparison serves the two purposes: (1) the Fourier-wavelet spectra are similar to those derived from the least-squares fitting method, and (2) the computation time for the Fourier-wavelet is much shorter than that for the least-squares fitting method. Accordingly, the text has been revised:

(lines 245-253): "Figure 1d is the same as Figure 1c but derived with the least-squares fitting method, which is often used for studying global-scale waves in the MLT region (e.g., Fan et al., 2022; Qin et al., 2022). The analysis was performed using time windows that are 3 times the wave period, which is a common choice in investigations of traveling planetary waves (e.g., Forbes and Zhang, 2015; Yamazaki and Matthias, 2019). The amplitude is not computed at the beginning and end of the data, where the length of the data is less than 3 times the wave period. There is good agreement between the results derived with the Fourier-wavelet (Figure 1c) and least-squares fitting (Figure 1d) methods. However, the computation time for the Fourier-wavelet method is approximately 1/100 that for the least-squares fitting method, highlighting the advantage of the Fourier-wavelet method in computation speed. Figures 1e−1g correspond to Figures 1b−1d but for wave_B."

Also, in Sec. 4:

(lines 333-336): "The Fourier-wavelet method has an advantage over other existing methods in that the computation is fast. For the example presented in Section 3.1, the computation time for the Fourier-wavelet method is approximately 1/100 that for the least-squares fitting method."

[2-3] 2. The authors can appropriately show the results on the planetary waves or tides in the ionospheric parameters (e.g., total electron content) extracted by the CFW transform.

[**Response**] For this paper, I would prefer to focus on global-scale waves in the MLT region. I am currently working on another manuscript describing the application of the Fourier-wavelet analysis to ionospheric data. In the revised manuscript, the applicability of technique to non-MLT data is addressed as future work.

(lines 337-339) "Although this study has focused on tides and traveling planetary waves in the MLT region, the Fourier-wavelet method can be easily applied to data from other regions of the atmosphere. Also, the applicability of the technique in research areas outside atmospheric science is yet to be explored."

---

## Author Response (AR2)

**Replies to Reviewers' comments on**
"A method to derive Fourier-wavelet spectra for the characterization of global-scale waves in the mesosphere and lower thermosphere, and its Matlab and Python software (fourierwavelet v1.1)"

I am thankful to the Reviewer, Dr. Jun-Ichi Yano, for taking the time to provide more comments. My response to individual comments can be found below. Like the last time, the comments from the reviewers are highlighted in blue, and the text from the revised manuscript is highlighted in red.
* * *
**RC1 (Dr. Jun-Ichi Yano)**

In the original review, I stated that "This is an interesting piece of work, potentially worthwhile for a publication". My basic position on this manuscript has not changed after the revision. Though I did not say this explicitly in initial review, I do not find any serious defect in this work. In this very respect, I do not wish to block the publication of this work in any fundamental manner. The only remaining question is the quality of the publication that both the Editor and the author wishes to achieve. My following comments intend to serve for this purpose:.

[**Response**] I am glad that the reviewer did not find any serious defect. It is my intention to deliver my work in its best shape. So, I truly appreciate that the reviewer is trying to help from his perspective.

Most seriously, the two major points made in my initial review are practically not at all taken into account in revision:

1) review of the existing methodologies

2) proper review of the wavelet method

1) more specifically, I proposed to refer to Zagar et al (2015, 2016). However, the authors rejects to do so by simply stating that the normal mode approach is not relevant in the present context. Most seriously, this remark is found only the response to the reviews, and very strangely, not found in the manuscript text. As a result, as it stands for now, the text reads like claiming that this Fourier-wavelet method is an only possibility of describing the propagating waves. Of course, the readers who are familiar with those existing methodologies would just wonder why one must use this when an existing method works well enough: what is an advantage of this proposed methodology against the existing methodologies?

[**Response**] I have revised manuscript to expand on the detection of normal modes in the troposphere and stratosphere. The Zagar et al. (2015) paper is cited therein:

(lines 34-39): "Global characteristics of normal modes can be predicted based on the linear wave theory (Kasahara and Puri, 1981; Zagar et al., 2015; Marques et al., 2020). Spectral analysis of meteorological data has confirmed the existence of waves similar to those theoretically predicted in the troposphere and stratosphere (e.g., Madden, 2007; Sakazaki and Hamilton, 2020). However,

characteristics of traveling planetary waves in the MLT region are expected to deviate considerably from those of theoretical normal modes due, for example, to dissipation and mean winds (Salby, 1981c)."

2) As it stands for now, there is even no proper lead sentence io introduce what the wavelet is. There is only a very short lead paragraph in Sec. 1.3, which is even misleading:

a) Torrence and Campo (1998) is just an introductory essay on wavelet. It would be misleading to state that the methodology of this paper is "described" in this essay: please cite a more proper textbook. my preference is one by Mallat.

[Response] In the revised manuscript, Sec. 1.1 introduces the wavelet technique citing the Mallat book.

(line 95-100): "The wavelet analysis (e.g., Mallat, 1999) is a multiresolution analysis technique using a 'wavelet', which is a short-term duration wave. A wavelet transform can be performed on one-dimensional (1-D) time-series to derive a 'wavelet spectrum', which is usually presented in a time versus period diagram. The wavelet spectrum is useful for identifying wave activity that is localized in time. The wavelet algorithm avoids the use of a moving window, which makes the technique more computationally efficient than the short-term analysis."

Sec. 1.3 gives a brief summary of the Torrence and Compo (1998) technique, providing the information that is required for understanding Sec. 1.4 (Fourier-wavelet analysis). It is not my intention to give a full description of the Torrence and Compo technique. This is now clarified in the revised manuscript.

(lines 145-148): "A wavelet analysis is performed in time. The method considered here is the continuous wavelet transform described by Torrence and Compo (1998). Their wavelet software including those in Matlab and Python are available from the website [http://atoc. colorado.edu/research/wavelets/]. The Torrence and Compo technique is widely used in atmospheric science due to its ease of use. Below the technique is only briefly summarized. Readers are referred to Torrence and Compo (1998) for full details of the technique."

b) As the authors states, with the continuous wavelet, a user can arbitrary select any frequency resolution, but only in a meaningless manner: this is just like trying to define the amplitude of noninteger wavenumbers over a finite domain, say, a wavenumber 1.345. Of course, one can do this, but we all know that this is meaningless. Strangely, the community simply does not realize the same with the continuous wavelet. The present author is not an exception. More formally stated, such an attempt contradicts with Heisenberg's uncertainty principle. Please refer to Mallat's text for the full discussions.

[Response] The text describing the advantage of the continuous wavelet transform over the discrete wavelet transform in Sec. 1.3 has been removed.

The advantage and applicability of the discrete wavelet transform is noted in the conclusion section (Sec. 4).

(lines 355-369): "Future work includes the improvement of the technique for faster computation and broader applications. The technique introduced in this paper relies on the 'continuous' wavelet transform. Studies have shown that the

'discrete' wavelet transform has some advantages such as non-redundancy (and hence more efficient computation) and straightforward invertibility (e.g., Mallat, 1999; Yano et al., 2001b, a, 2004). The discrete wavelet transform may be implemented in the Fourier-wavelet technique.

An important limitation of the Fourier-wavelet technique is that it can resolve only global-scale waves. Along with tides and traveling planetary waves, gravity waves are also important in the MLT region (e.g., Fritts and Alexander, 2003; Smith, 2012), with a wide range of zonal wavenumbers (up to 100 or so) (e.g., Miyoshi and Fujiwara, 2008; Liu et al., 2014). Since gravity waves are often localized in space, the Fourier-wavelet technique would not be able to fully capture them. A 2-D wavelet analysis (e.g., Kikuchi and Wang, 2010) would be useful. An easy-to-implement 'wavelet-wavelet' technique for evaluating gravity-wave amplitudes and phases may be developed as an extension of the Fourier-wavelet technique presented in this paper.

Although this study has focused on waves in the MLT region, the Fourier-wavelet method could be applied to data from other regions of the atmosphere. The technique may also be useful in research areas outside atmospheric science. The extent of applicability of the technique is still to be explored."

c) In short, there is no real advantage with the continuous wavelet against the discrete counterpart, apart from an easiness of using it. This very last point should be made absolutely clear in a final version with more relevant references (including a textbook by Mallat).

[**Response**] In the revised manuscript, the text describing the advantage of the continuous wavelet transform has been removed, and it is stated that the Torrence and Campo (1998) technique has been widely used because of its ease of use.

(lines 145-148): "The method considered here is the continuous wavelet transform described by Torrence and Compo (1998). Their wavelet software including those in Matlab and Python are available from the website [http://atoc. colorado.edu/research/wavelets/]. The Torrence and Compo technique is widely used in atmospheric science due to its ease of use."

The Mallat reference is cited in Sec. 1.1 when the wavelet technique is mentioned for the first time.

3) The introduction as it stands for now, mostly consists of a review of the existing studies on the plain planetary waves. The given review hardly motivates the present study.

In this context, the author totally neglects the following full paragraph from my initial review:

"In this respect, the introduction is slightly confused as it stands for now: its first half reviews previous works detecting "linear" planetary waves. Then, suddenly, at L55, the author decides to talk about the stratospheric sudden warming: this is clearly a nonlinear process that cannot be described by a single wave. The authors further begins to remark that the observed waves are rather "intermittent" (in own wording), and they can emerge even like bursts: that is all fine with me: these observed waves are not perfectly linear, and they are often

generated by forcings as well as instabilities, and those evolution can be very nonlinear. However, after said all those (though the author does not comment on them), if one wishes to understand those phenomena as a part of the wave dynamics, an obvious way to go is to perform the normal-mode decompositions so that one can see explicitly which modes are involved in processes in which manner, etc. Those are very basic backgrounds of the atmospheric-wave dynamics, that should remind the readers."

[Response] Traveling planetary waves in the MLT region usually do not behave like normal modes of the linear wave theory. That is the reason the normal-mode decomposition function technique has limited applicability there. I have re-emphasized this point in the revised manuscript.

(lines 37-40): "However, characteristics of traveling planetary waves in the MLT region are expected to deviate considerably from those of theoretical normal modes due, for example, to dissipation and mean winds (Salby, 1981c). Also, some traveling planetary waves in the MLT region are considered to be unstable modes locally generated by atmospheric instability, rather than normal modes (e.g., Pfister, 1985; Meyer and Forbes, 1997)."

(lines 45-46): "The zonal wavenumber and wave period of these waves are consistent with Rossby modes of the linear wave theory, but their meridional and vertical structures are generally different from those of theoretical Rossby modes."

(lines 54-56): "Observations also sometimes show westward-propagating planetary waves in the MLT region whose periods do not match those of normal modes (e.g., Qin et al., 2022a, 2021b)."

An only hint for a need for the wavelet is a short phrase of "a burst of wave activity" (L57). However, without any proper elaboration, it is even not clear what the author is exactly referring to. Finally, at L73, the author states, "This is the motivation.....". However, unfortunately, I cannot identify a sentence that can be called a motivation in any earlier part: then what is this "This"? The author quickly adds a well known fact that "The wavelet analysis is useful for identifying wave activity that is localized in time" (L79): however, how often we observe such isolated waves in the atmosphere?etc The author simply fails even to provide such basic information.

[Response] In the revised manuscript, I have elaborated on the transient nature of global-scale waves in the MLT region.

(lines 64-69): "Traveling planetary waves in the MLT region sometimes show a burst of wave activity that lasts for a few wave cycles. This can be attributed to changes in the zonal mean state of the atmosphere, which controls propagation conditions, atmospheric instability, and critical layers (e.g., Salby, 1981b, c; Liu et al., 2004; Yue et al., 2012; Gan et al., 2018). A wave burst is often observed around seasonal transition, but its characteristics (e.g., magnitude, peak period, meridional structure, and so on) vary from year to year, so that it is difficult to predict them (e.g., Gu et al., 2019; Liu et al., 2019; Yamazaki et al., 2021). Also, some wave burst events occur during sudden stratospheric warmings."

The logic of the presentation is very loose at the best at many occasions. The most notable example is found in L80-81, which states: "the standard 1-D

wavelet technique is not directly applicable to two-dimensional (2-D) longitude-time data...." If I understood the author's response to my comment correctly, this is nothing other than a trivial statement that any 1D transformation technique (wavelet or Fourier or else) is not directly applicable to any 2D data, because the transformation method is only 1d, and not 2D. Thus, we must adopt either 2D wavelet or 1D Fourier technique. It does not follow at all the we need to invoke Hayashi's method to overcome this difficulty. We just need to invoke any available 2D techniques.

[**Response**] The manuscript has been revised to make the motivation clearer and more logical.

(lines 86-104): "Characterization of global-scale waves requires the identification of the zonal wavenumber and wave period (see equation 1), which demands two-dimensional (2-D) spatiotemporal data (more specifically, data as a function of longitude and time). Techniques such as 2-D fast Fourier transform (FFT) (e.g., Hayashi, 1971) and 2-D least-squares fitting method (e.g., Wu et al., 1995) can be applied to the data to evaluate the zonal wavenumber and wave period of global-scale waves and their amplitudes and phases. Taking into account the transient nature of global-scale waves in the MLT region, a short-term analysis is commonly used. That is, a 2-D spectral analysis is performed on a short-time segment of the data, then the analysis window is moved forward in time (e.g., Maute, 2017; Forbes et al., 2018; Liu et al., 2021). This way, it is possible to evaluate temporal variations of global-scale waves. However, such a moving-window approach is computationally expensive because the spectral analysis needs to be repeated for multiple times. As a solution to this problem, this study proposes the application of wavelet analysis. The wavelet analysis is a multiresolution analysis technique (e.g., Mallat, 1999). A wavelet transform can be performed on one-dimensional (1-D) time-series to derive a 'wavelet spectrum', which is usually presented in a time versus period diagram. The wavelet spectrum is useful for identifying wave activity that is localized in time. The wavelet algorithm avoids the use of a moving window, which makes the technique more computationally efficient than the short-term analysis. The main objectives of this study are (1) to introduce a simple method to derive 'wavelet-like' spectra from 2-D longitude-time data, which can be used for the characterization of global-scale waves in the MLT region, and (2) to deliver easy-to-use software in two user-friendly languages: Matlab and Python. For (1), the 2-D FFT method of Hayashi (1971) is used, and it is modified by adopting the wavelet technique of Torrence and Compo (1998). The Hayashi (1971) method is easy-to-implement and its spectrum directly gives the wave amplitude in units of the input data, which is easy to interpret."

In conclusion, I see that still substantial revisions are required before this manuscript becomes worthwhile for publication. [I click "major revision" because more than a cosmetic modification is required: the author clearly failed to note this point in then initial revision] As the very minimum, misunderstanding concerning the wavelet by the author must definitely be corrected.

Thank you again for a guideline for improving the manuscript.

More Specific:

L77: please add a reference discussing on the discrete wavelet in a more proper manner, for example: Yano et al (2001a, b)

Here I cite the Mallat book. But Yano et al. (2001a, b) are cited in Sec. 4 in addressing the future work.

L83, "Hayashi's method is combined with the wavelet technique": probably is would be more proper to say that "Hayashi's method is modified by adopting the wavelet in representation in time."

The change has been made as suggested.

(line 102): "For (1), the 2-D FFT method of Hayashi (1971) is used, and it is modified by adopting the wavelet technique of Torrence and Compo (1998)."

L130, Yano et al (2001, 2004): please also to refer to the second part (Yano 2001b) for a completeness.

The change has been made as suggested.

(lines 356-358): "Studies have shown that the 'discrete' wavelet trans- form has some advantages such as non-redundancy (and hence more efficient computation) and straightforward invertibility (e.g., Mallat, 1999; Yano et al., 2001b, a, 2004)."

---

## Author Response (AR3)

**Replies to Reviewers' comments on**
**"A method to derive Fourier-wavelet spectra for the characterization of global-scale waves in the mesosphere and lower thermosphere, and its Matlab and Python software (fourierwavelet v1.1)"**

I am, again, thankful to the Reviewer, Dr. Jun-Ichi Yano, for the time and valuable comments. Please find below my response to individual comments.
* * *
**RC1 (Dr. Jun-Ichi Yano)**

[instead of] adding the materials on top of the existing materials, [better to try] to tide up the text to make the main points straight. [...] the text, especially the introduction has become extremely verbose, full of very secondary, removable remarks.

[Response] I understand that the reviewer would write the introduction differently. However, I would like to keep the contents of the current introduction. For me, the current introduction contains all the information that is necessary to understand the subsequent sections. I could make it longer but it is difficult to make it shorter without losing some important information.

[it is essential] to state explicitly that the continuous wavelet has a fundamental limitation of providing more information than what are actually available under the Heisenberg's uncertainty principle.
[lack of it in the text is] extremely unfortunate, because this manuscript will just continue to promote existing very limited use of the wavelet methods in our community.

[Response] As suggested, it is explicitly stated in the revised manuscript that the continuous wavelet has an inherent problem. The following sentence has been added:

(lines 358-360) "One criticism against the continuous wavelet transform is that it provides more information than what is actually available under the Heisenberg's uncertainty principle (e.g., Yano and Jakubiak, 2016)."

[...] Torrence and Compo (1998) did not propose any original methodology for the wavelet analyses. [last revisions] introduce a new term "Torrence and Compo" technique. However, I was left puzzled what this technique is about. I believe that Mallat's textbook describes the continuous wavelet method more systematically than Torrence and Compo (1998). [...]

[Response] By "the Torrence and Compo technique is widely used", I meant their wavelet software. I admit that the text was not clear. So, in the revised manuscript, it has been made clear that I am talking about their wavelet software. The following change has been made:

**From**

"Their wavelet software including those in Matlab and Python are available from the website [http://atoc. colorado.edu/research/wavelets/]. The Torrence and Compo technique is widely used in atmospheric science due to its ease of use."

**To**

(lines 147-149) "Their wavelet software including those in Matlab and Python are available from the website [http://atoc.colorado.edu/research/wavelets/], which are widely used in atmospheric science due to its ease of use."